# Multifaceted Role of Notch Signaling in Vascular Health and Diseases

**DOI:** 10.3390/biomedicines13040837

**Published:** 2025-03-31

**Authors:** Ahsan Ali, Sanguk Yun

**Affiliations:** Department of Biotechnology, Inje University, Gimhae 50834, Republic of Korea; ahsanarif125@gmail.com

**Keywords:** vasculature, endothelium, notch, cell signaling, inflammation, mechanotransduction

## Abstract

Notch signaling is evolutionarily conserved from *Drosophila* to mammals and it functions as an essential modulator of vascular growth and development by directing endothelial cell specification, proliferation, migration, arteriovenous differentiation, inflammation, and apoptosis. The interplay between Notch and other signaling pathways plays a homeostatic role by modulating multiple vascular functions, including permeability regulation, angiogenesis, and vascular remodeling. This review explores current knowledge on Notch signaling in vascular development, homeostasis, and disease. It also discusses recent developments in understanding how endothelial Notch signaling affects vascular inflammation via cytokines or aberrant shear stress in endothelial cells while addressing the reciprocal relationship between Notch signaling and inflammation.

## 1. Introduction

The vasculature, which is characterized by diverse types of blood vessels ranging from microscopic capillaries to large vessels, is not merely a network of blood vessels that nourishes tissues and organs, but it also contributes to the regulation of tissue growth, homeostasis, and repair [1,2,3]. To achieve these complex tasks, the number, size, and hierarchy of the vessels must be precisely regulated. Notch signaling regulates the volume and organization of vascular cells by controlling interactions between different cell types within the vasculature. It modulates the differentiation of endothelial and smooth muscle cells (SMCs) to ensure appropriate vessel emergence and homeostasis during development [4].

The developmental process is dependent on complex cell-cell interactions, from a single cell to a diversified multicellular organism. These interactions are vital for determining cell fate and the formation of biological patterns. Because of the key functions of Notch receptor and related pathway components in development, human diseases characterized by aberrant cellular differentiation and development are frequently associated with mutations in members of this pathway. The disruption of cell-cell interactions is linked to many genetic disorders, such as Alagille syndrome and cerebral autosomal dominant arteriopathy with subcortical infarcts and leukoencephalopathy (CADASIL) [5,6].

Inflammation is a complex and dynamic process involving a diverse array of cells and signaling components triggered by various tissue-remodeling stimuli [7]. Notch signaling and inflammatory responses are closely related. This association is primarily driven by immune cells, which utilize the Notch pathway to modulate the inflammatory process [8]. Macrophages are crucial for maintaining homeostasis, immune surveillance, and managing inflammation in almost every tissue. Macrophage activation, polarization, and differentiation are influenced by Notch signaling [9]. Numerous cell types, including ECs, SMCs, and macrophages, exhibit pro-inflammatory phenotypes when triggered by Notch activation. This activation exacerbates inflammation and accelerates the development of atherosclerosis and other inflammatory disorders [10].

A key component of vascular system development is fluid shear stress, which is a frictional force exerted by blood flow on the vessel wall [11]. A decrease in shear stress because of reduced blood flow or viscosity results in improper vascular development and remodeling [12]. Notch signaling and hemodynamics are closely related [13]. Notch functions as a mechanosensor of shear stress and mediates stem cell differentiation into ECs, as well as vascular development [14,15,16,17].

### Overview of Notch Signaling

Notch signaling is a contact-dependent pathway involved in multiple cellular processes, such as cell fate determination, development, proliferation, inflammation, homeostasis, and apoptosis. Mammals have four distinct Notch receptors (Notch1–4) and five ligands (JAG1-2 and DLL1, 3, and 4). These are all type I transmembrane proteins, and their activation requires cell-to-cell contact. Generally, interactions between Notch ligands and receptors occur in neighboring cells (either of the same type [homotypic] or different types [heterotypic]), leading to trans-signaling events. However, these ligands can also bind to receptors in cis within the same cell plasma membrane [18,19]. The long Notch extracellular domain (NECD) contains a series of N-terminal epidermal growth factor (EGF)-like repeats and a juxtamembrane negative regulatory region (NRR) which includes three Lin12/Notch repeats (LNRs) and a heterodimerization domain [20,21]. The Notch intracellular region contains a RAM domain for protein binding, an ankyrin repeat domain (ANK) and a PEST domain [22]. The complex molecular structure of both Notch receptors and Notch ligands is shown in Figure 1.

The two known mechanisms by which Notch exerts its functions have been debated for several decades. The CBF1/RBP-j/Su(H)/Lag-Delta and Notch-like epidermal growth factor-related receptor (CSL)-dependent canonical Notch pathway is more comprehensively defined than the non-canonical pathway. It is triggered when one of the Notch receptors binds to a ligand, causing conformational changes in the receptor, resulting in a cascade of proteolytic cleavage events. The first proteolytic event is governed by ADAM-family proteases (ADAM10, ADAM17, and ADAMTS1), resulting in the shedding of the NECD from the cell membrane, followed by a second cleavage in the transmembrane domain catalyzed by a γ-secretase complex, which releases the active Notch intracellular domain (NICD). NICD translocates to the nucleus and binds to the CSL transcription factor (CBF1/RBP-Jκ in mammals), converting a repressor form to an activated form. NICD facilitates the recruitment of RBP-Jκ and histone acetyltransferases, resulting in the activation of mastermind-like (MAML) to drive the transcription of primary notch target genes (*Hes1* and *Hey*) [23,24]. The RBPJκ/CSL-independent non-canonical pathway can be activated by a non-canonical ligand and does not always require Notch receptor cleavage to occur. Usually, interactions with CSL and activation of *Hes*/*Hey* gene expression are circumvented in this pathway, and target gene regulation occurs through distinct mechanisms. NICD interacts with the P13K/AKT, NFκB, Wnt, HIF-1α, or mTORC2 pathway to regulate cell viability, metabolism, and differentiation [25]. In addition, a newly discovered mechanism of non-canonical activation has emerged. Traditionally, S3 cleavage of Notch has been considered essential for the release of the Notch Intracellular Domain (NICD) and the subsequent regulation of target gene transcription. However, it has been found that membrane-associated Notch can activate the PI3K-AKT pathway to increase the transcription levels of *IL-10* and *IL-12* [26].

## 2. Notch and Vascular Development

### 2.1. Vasculogenesis

At the onset of embryonic development, the vascular system must develop de novo through vasculogenesis [27]. Originally derived from mesodermal cells, endothelial progenitor cells, also known as angioblasts, form an early vascular network (plexus) supported by mural cells that undergo maturation and remodeling into arteries, veins, and capillaries starting at approximately E7.5 in mouse embryos [28]. The transition of angioblasts into ECs is regulated by a complex array of transcription factors and signaling pathways [29,30,31]. In tandem with this, extraembryonic vasculogenesis occurs in the yolk sac, where ECs arise from hemangioblasts, which play a dual role in the creation of blood vessels and the formation of hematopoietic cells [32]. By E8, angioblasts begin to form the first distinct vascular structure through the formation of a primitive paired dorsal aorta [33]. Genetic deletion of Notch components, such as Jagged1, Notch1, Notch4, Hey1, and Hes1, in mice results in embryonic lethality, indicating that appropriate vascular network formation and maturation are not possible without the Notch pathway [34,35,36]. In a similar way, restricting the expression of CSL or Notch1 in ECs led to severe vascular phenotypic defects [35,37,38].

Notch signaling is pivotal in controlling arteriovenous fate specification, and its importance in arterial differentiation was initially demonstrated in zebrafish and later in transgenic mice and human ECs [39,40,41,42,43]. Dysregulated Notch signaling results in reduced arterial endothelial differentiation and increased venous differentiation. Notch signaling regulates arterialization via epistatic regulation of Eph/ephrin signaling [44]. In the early vasculature, Notch induces ephrinB2 expression and inhibits EphB4 expression [38,45,46]. In ECs, vascular endothelial growth factor (VEGF) plays a key role in directing the design of the vascular blueprint by upregulating the Dll4 ligand and Notch 1 receptor in the arterial region of the vasculature, thereby promoting arterial fate. On the venous side of the vasculature, COUP-TFII inhibits Notch signaling, which ultimately leads to the venous fate [47]. VEGF is a key regulator of the blood vessel network via chemotactic signals [48]. Notch1 in VEGFA-deficient embryos restores EphrinB2 expression; however, VEGFA delivery into Notch-pathway-deficient embryos fails to restore arterial differentiation in zebrafish [39]. Liu et al. observed that VEGFA induces Notch1 and Dll4 expression in human arterial ECs (HAECs) [49]. In mouse embryos, VEGFA activation induces Notch4, Dll4, and EphrinB2 while suppressing EphB4 and COUP-TFII [50].

### 2.2. Angiogenesis

Angiogenesis is defined as the formation of new blood vessels from pre-existing basic vascular networks to promote ongoing development [51]. Angiogenesis is influenced by biochemical and mechanical signals [52]. During the early stages of angiogenesis, fibroblast growth factors (FGFs) and VEGF promote EC growth and the emergence of new vascular networks [48]. One of the key events in sprouting angiogenesis is the differentiation of ECs into two specialized cells known as tip and stalk cells. Tip cells guide new vessel sprouts by working with other signaling pathways, such as the VEGF-A/VEGFR2 pathway, and stalk cells give rise to the lumen of the new vessel [53]. Functional studies have shown that Notch signaling operates along with the VEGF pathway in controlling angiogenesis [54]. Endothelial proliferation and cell cycle regulation depend on distinct levels of VEGFR and Notch signaling [55,56,57]. VEGF stimulation activates the ERK pathway, which enhances the production of Dll4 in the future tip cells [58]. VEGFR signaling is strong in ECs positioned at the leading edge of growing blood vessels [59]. ECs at the angiogenic front show low levels of Notch signaling compared to adjacent stalk cells expressing JAG1 when exposed to elevated VEGF-A levels. As a result, tip cells experience the strongest mitogenic response caused by elevated levels of VEGFR2 signaling and low levels of Notch signaling, resulting in peak activation of MAPK/ERK signaling [60,61]. DLL4, which is expressed in tip cells, activates Notch signaling in stalk cells, causing pERK levels to be lower than those in tip cells, preventing p21-mediated cell cycle arrest. Therefore, stalk cells are the most proliferative cells because of Notch signaling and continued exposure to VEGF-A [57]. A recent study indicated that Hey1 controls angiogenesis by regulating positive (FGF and VEGF pathways) and negative (Notch pathway) signals through the dynamic regulation of SUMOylation and deSUMOylation. Physiologically, SUMOylation enhances the transition of Hey1 homodimers to Hey1-Hes1 heterodimers, and pro-angiogenic stimuli cause deSUMOylation. A deficiency in SUMOylation changes the DNA-binding capacity of Hey1 and reduces the transcriptional suppression of target genes, including RTKs and Notch components [62]. Recent findings by Duan et al. indicated that in a mouse retinal model, angiogenesis is regulated by oxygen sensing and Notch signaling in ECs [63]. In Phd2^EC−/−^ mice, increased HIF2α accumulation impairs angiogenesis, even with high levels of VEGF-A. Phd2^EC−/−^ mice exhibited unexpectedly high levels of DLL4 and Hey2 in their INL/IPL regions. Future research should investigate whether PHD2-deficient retinal ECs produce factors that promote DLL4 production in these tissues, as the latest preprint has revealed that DLL4 can be distributed extracellularly [64]. Different strains have been used to study the effect of Notch signaling in mouse models as shown in Table 1.

### 2.3. Notch and the Lymphatic Vasculature

In addition to the blood vasculature, mammals have another type of vasculature known as the lymphatic vasculature [78]. Lymphatics are the primary vessels of the lymphatic system, and they are formed by a single layer of lymphatic ECs that mediate the absorption of macromolecules and interstitial fluid [79]. The lymphatic microvasculature is specifically designed to facilitate the removal of proteins and interstitial fluid [80]. The lymphatic system performs various immune functions, including antigen import, antigen presentation, immune cell trafficking, regulation of immune cell activation, and promotion of survival [81]. Angiogenesis of blood vessels is driven by VEGFR2 in endothelial tip cells responding to VEGFA gradients [53,82], lymphangiogenesis, the formation of new lymphatic sprouting from pre-existing ones, which depend on VEGFR3 responding to VEGF-C [83,84,85,86]. The mesenchymal tissue near the cardinal vein secretes VEGF-C, which interacts with VEGFR3, triggering a series of crucial cellular responses including sprouting, migration, proliferation, and survival of LECs (lymphatic endothelial cells) [83,85,87]. Lymphatic vessels are fundamentally different from blood vessels in many aspects. For instance, lymphatic vessels are loosely connected to gaps and lack pericytes around lymphatic capillaries [88,89]. Notch1 is a key regulator of the maintenance of the lymphatic lineage and the prevention of venous specification [90]. During the maturation of the lymphatic vasculature, Notch1-Dll4 regulates the sprouting of lymphatic vessels during postnatal lymphangiogenesis [67]. It is not surprising that blocking Notch signaling with specific antibodies lowers the number of lymphatic vessels [91]. In contrast to the arterial endothelium, where laminar fluid flow upregulates Notch signaling, laminar flow in lymphatic vessels downregulates Notch signaling and thus, promotes lymphatic endothelial sprouting [91,92].

### 2.4. Notch Signaling and Mural Cells in Vascular Design

Effective blood vessel formation and functional maturation require complex coordination and cellular interactions between ECs and surrounding mural cells. Many signaling molecules and pathways have been found to affect mural cell coverage during vascular development [93]. In the arterial compartment of blood vessels, Notch signaling is vital and well-regulated for proper functioning and organization of vascular SMCs [94,95,96,97]. Coordination between JAG1 on ECs and Notch receptors on mural cells enables SMCs to gain contractile properties required for vascular function [98,99,100]. This coordination also plays a role in SMC adhesion to the basement membrane and in maintaining endothelial extracellular matrix homeostasis [72,101,102]. JAG1 is a downstream target of Notch signaling in SMCs, which leads to spreading waves of JAG1-Notch signaling through the layers of smooth muscles of larger arteries, thereby promoting the assembly and structural integrity of the arterial wall [103,104]. There is evidence that Jag1 and Notch3 are the sole ligands and receptors involved in the Notch signaling pathway required for the SMC lineage [105]. Notch3 expression is selectively found in vascular smooth muscle cells (VSMCS) rather than veins [94]. Jag1 primarily promotes the differentiation of adjacent VSMCs [99]. In pericytes, the Notch pathway directly regulates the levels of PDGFRβ and Notch disruption causes a marked depletion of pericytes in the zebrafish brain and the vascular plexus in the mouse retina [69,101,106,107]. Notably, endothelial JAG1 deletion does not modify pericyte coverage, indicating a different function of Notch signaling in pericytes than in SMCs. [72,108] Moreover, little is known whether Notch signaling between pericytes has any biological importance [4]. Future studies may uncover the complex interactions between endothelial and mural cells that shape the structure and specialization of blood vessels.

## 3. Notch and Vascular Inflammation

ECs, SMCs, fibroblasts, and immune cells, such as macrophages and dendritic cells, play critical roles in the regulation of inflammation in blood vessels. Notch receptors and their ligands exhibit distinct cell-type-specific expression and activity, which is supported by the activation of downstream target genes (*Hes* and *Hey*) and evidence from CBF-1 reporter assays [109,110]. Soluble inflammatory mediators, such as TNF-α, IL1β, and IFNγ, alter the expression levels of Notch receptors in ECs. TNF-α and IL-1β significantly downregulate Notch4 expression levels, while upregulating Notch2, Jag1, and Dll1 expression levels, whereas IFNγ has no effect on Notch4, highlighting a unique regulatory mechanism for each Notch receptor during inflammation [111].

### 3.1. Notch and Endothelial Inflammation

ECs respond to various inflammatory stimuli and promote monocyte recruitment and transmigration via upregulation of monocyte adhesion molecules and the induction of endothelial permeability [112]. Notch is implicated in isoform-specific endothelial inflammation. The function of Notch1 in endothelial inflammation is highly context-dependent. The homeostatic function of Notch1 has been demonstrated in animal models of atherosclerosis and pulmonary arterial hypertension [113,114]. However, there are also reports showing that blockade of Notch1 reduces cytokine-induced endothelial inflammation and that endothelial Notch1 knockout leads to the suppression of atherosclerosis [115,116]. Opposing Notch1 functions in vascular permeability have previously been reported. Notch1 promotes endothelial barrier function via canonical and non-canonical mechanisms [117,118]. However, Notch1 blockade under diabetic conditions or DLL4-bound Notch1 enhances barrier integrity [119,120]. TNF-α induces Notch2 in ECs, but the functions of Notch 2 and Notch 3 in endothelial inflammation have not yet been revealed [111]. Notch4 mediates shear-stress-induced endothelial inflammation in atherosclerosis and varicose veins [121,122]. Notch ligands that regulate endothelial inflammatory phenotypes have also been identified. DLL1 weakens endothelial barrier function and Jag1 promotes inflammatory leukocyte recruitment and atherosclerosis [66,70,116]. Knockdown of Jag1 or Dll4 leads to the suppression of IL-1b-induced VCAM-1 expression in ECs [115]. Dll4 blockade also reduces endothelial proliferation and tumor angiogenesis in various tumor models [123]. TNF-α induces Jag1 expression in human umbilical vein ECs and the plating of ECs on Jag-coated surfaces decreases inflammatory gene expression levels [124]. Endothelial Jag2 promotes pulmonary arterial hypertension by upregulating GATA transcription factors [125]. Certain Notch receptors and ligands contribute to endothelial inflammation, vascular permeability and atherosclerosis, while others have neutral role as shown in Table 2. 

Endothelial senescence significantly contributes to vascular dysfunction and the progression of age-related diseases [68,126]. Endothelial Sirt1 loss triggers senescence, fibrosis, vascular permeability and inflammation [127,128]. Sirt1 deletion causes a significant stimulation of Notch signaling, as evidenced by the upregulation of Notch target genes, such as *Hey1* and *Hes1*, *DLL4*, and *NICD*, ultimately causing endothelial dysfunction [129]. Sirt1 overexpression in SMCs increases Notch signaling to prevent vascular aging [127].

### 3.2. Notch and Inflammatory Signaling Crosstalk

#### 3.2.1. Notch and the NFκB Pathway

Both the Notch and NF-κB signaling pathways are key regulators of various vital cellular processes, including proliferation, differentiation, and apoptosis. Although distinct in their mechanisms, these pathways often interact and overlap during developmental regulation and immune responses [130]. In different experimental models, the NF-κB pathway controls inflammation in ECs and immune cells, such as macrophages, in a variety of ways [131]. Various biological assays indicate that Notch1 directly collaborates with NF-κB p65, promoting NF-κB pathway activity and the expression of other inflammatory cytokines, such as TNF-α and IL-1β [132]. Inhibiting γ-secretase or silencing Notch1 regulates the translocation of p50 to the nucleus under lipopolysaccharide (LPS)/IFNγ stimulation [133]. The Notch1 intracellular domain prevents the transcriptional activity of NF-κB by interacting with p50, thereby blocking DNA binding. CBF-1 protein binds to the *IκBα* promoter region repressing IκBα expression and inducing NF-κB activation. In addition, p65 sequestration by IκBα causes SMRT/N-CoR to translocate to the cytosol, enhancing the expression of Notch target genes [134,135,136]. A recent study showed that LPS-activated Notch3 and Notch1 are crucial for controlling NF-κB activity in macrophages by increasing p38 activation [137]. Moreover, Notch1 signaling influences NF-κB activation by regulating the CYLD-TRAF6-IKK pathway activated by TLR-4 signaling, thus increasing the expression levels of pro-inflammatory cytokines, such as TNF-α, IL-1β, and IL-6 [138]. Apurinic/apyrimidinic endonuclease (APE1) is activated in cancerous tissues and activates redox-regulated transcription factors [139,140]. Recently, Chan et al. found that under reflux conditions, DLL1 activates the Notch1 receptor, which is driven by the NF-κB pathway, showing an important relationship between reflux-driven inflammation and the redox function of APE1 and Notch signaling in esophageal adenocarcinoma cells [140]. The APE1-NF-κB-Notch1 signaling pathway is essential for tumor initiation and progression [141]. Two proteins belonging to NF-κB family, p52 and RELB, activate the Notch signaling pathway upon TNF-α stimulation [142]. When these proteins are overexpressed in mouse stem cells, Hes1 levels are upregulated [142]. Notch signaling interacts with other multiple pathways influencing vascular function as shown in Table 3.

#### 3.2.2. Notch and the Hypoxia Pathway

Notch-hypoxia interactions have recently been discovered in various cell types and physiological situations. Hypoxia impairs differentiation in various stem cell populations [145]. Hypoxia promotes stem cell maintenance and inhibits the differentiation of myogenic and neural progenitors via Notch signaling [146]. Low oxygen levels in pancreatic beta cells promote dedifferentiation through Notch1 signaling [147]. By elevating Notch1 intracellular domain levels, hypoxia hinders the osteogenic differentiation of mesenchymal stem cells, because the NICD binds to Runt-related transcription factor 2 and reduces its transcriptional activity [148]. Hypoxia affects ciliated cell development in the conducting airways by lowering the expression levels of forkhead box J1, which is mediated by Notch signaling, because DAPT suppresses Notch and restores differentiation [143]. HIF2α-deficient pancreases exhibit poor branching and endocrine distinction because of impaired Notch signaling. A direct interaction between Notch and HIF2α results in a reduction in the number of Hes1^+^ cells [149]. Both HIF1α and HIF2α promote the expression of Hey1, Hey2, and Dll4 in response to low oxygen conditions [65]. Hypoxia stimulates Dll4 via the hormone adrenomedullin during endothelium development from embryonic stem cells in a Notch-independent manner, and this stimulation is sustained by Notch signaling, resulting in a forward feedback loop [150]. HIF-1α stimulates NICD-dependent gene transcription by binding to Notch-responsive promoters in response to hypoxia [146]. ECs also respond to hypoxia by upregulating the Dll4 ligand, causing neighboring cells to activate Notch signaling [65]. Gene expression in the Notch cascade is favorably modulated by a low oxygen demand in the hypoxic intervertebral disc, which is the junction between the vertebrae. Therefore, disc tissue exhibits hypoxia-induced expression of Hes1, Jagged1, Jagged2, Notch1, and Notch4, as well as elevated levels of Notch 2 [151]. Sima, the ortholog of HIF1α in *Drosophila*, has been shown to ligand-independently activate Notch receptors, facilitating the growth of platelet-like crystal cells [152]. Additionally, the ability of *Drosophila* to adjust to and endure hypoxic stress has been reported to be a polygenic trait, indicating that the Notch pathway is essential for hypoxic adaptation [153]. Although the contribution of Notch to angiogenesis under hypoxic conditions is well documented, its relevance to hypoxia-induced inflammatory processes remains unknown.

## 4. Notch Signaling and Shear Stress

### 4.1. Mechanotransduction and Notch Signaling

Mechanotransduction refers to the conversion of mechanical stimuli into biochemical signals that allow cells to detect and respond to changes in their surrounding environment [154]. ECs, because of their direct close contact with the circulating blood, are influenced by shear stress. Ion channels are vital for the detection of shear stress. Piezo and transient receptor potential (TRP) channels are the most well-studied of these channels. Of note, recent data suggest that they may work in coordination. Caolo et al. identified a relationship between Piezo1 channels and Notch1 signaling [155]. Piezo1 activation induces ADAM10 and γ-secretase to release NICD, promoting Notch1-regulated expression of *HES1*, *HEY1*, and *DLL4* genes. Endothelial Piezo1 deletion impairs Notch1 signaling in hepatic ECs. The mechanical forces sensed by Piezo and TRP channels in both ECs and SMCs are influenced by Notch signaling during cardiac development. Piezo-mediated modulation of Klf2α expression in ECs influences Notch activation, whereas Piezo channels regulate Yap1 localization [156]. In mice, another mechanoreceptor, Pannexin-1, works upstream of TRPV4 and promotes ATP efflux. It interacts with TRPV4 and P2Y2R in Caveolin-1-rich lipid zones to increase eNOS activity and promote vasodilation, while lowering pulmonary arterial pressure [157]. Furthermore, Piezo-1 and Piezo-2 have been linked to mechanosensing in the pulmonary endothelium, with Piezo-2 specifically recognized for its direct response to shear stress. Piezo-2 is recognized for its immediate response to shear stress, and both Piezo-1 and Piezo-2 are linked to mechanosensing in the pulmonary endothelium [158,159]. Liu et al. demonstrated that Piezo1 in osteocytes regulates two key proteins, OPG and RANKL, in response to shear stress [160]. Piezo1 stimulates OPG expression and inhibits RANKL expression via a NOTCH3-dependent mechanism. Research on the impact of pharmacological inhibition of Piezo channels in cell cultures and mouse models of atherosclerosis has increased our understanding of their roles in shear stress signal transduction [161]. However, how Piezo receptors sense shear stress remains unknown.

### 4.2. Shear Stress and Arterial Venous Differentiation

Hemodynamic forces, such as shear stress, are essential for blood vessel homeostasis. Laminar flow, characterized by a unidirectional and steady flow, supports endothelial homeostasis, reduces inflammation, and promotes the release of nitric oxide (NO), thus inducing smooth muscle relaxation [162]. By contrast, disturbed shear stress promotes endothelial dysfunction, enhances inflammation, and reduces NO production, leading to atherosclerosis [163,164]. Notch signaling is important for maintaining arterial homeostasis in response to changes in hemodynamic flow. Notch1 functions as a mechanosensor in this regard [165]. Recent findings indicate that the modulation of shear stress during pluripotent stem cell differentiation into arterial ECs is mediated by VEGFR-Notch signaling [166]. The interaction between VEGF and its receptors, VEGFR1 (Flt-1) and VEGFR2 (Flk-1 and KDR), stimulates the endothelial differentiation of stem cells [167,168]. VEGFR sensitivity to shear stress, which occurs without ligand interactions, stimulates the Notch signaling cascade and plays a vital role in EC development [169,170]. Shear stress stimulates EC differentiation through the Notch-VEGFR-EphrinB2 signaling pathway. In a recent study of murine-ESC-derived VEGFR2^+^ cells subjected to shear stress (10 dynes/cm^2^ for 24 h), EphrinB2 upregulation was blocked by DAPT. Furthermore, the VEGFR kinase inhibitor SU1498 (10,000 mol/L) significantly reduces shear-stress-induced Notch cleavage, which is essential for the transactivation of promoters involved in embryonic vascular development.

### 4.3. Shear Stress and Cerebral Arteriovenous Malformations

Cerebral arteriovenous malformations (cAVMs) are rare vascular anomalies in which proliferative arteries and draining veins are directly connected, bypassing the capillary network and resulting in elevated flow and minimal resistance to arteriovenous shunting [171]. Previous studies have indicated that either activation or inhibition of Notch signaling can lead to the development of AVM [37,41]. High expression levels of Notch ligands, such as Dll4 and Notch target genes, including Hey2 and Ephrin B2, have been reported in the AVM nidus [172]. The initial activation of canonical Notch signaling is indicated by increased expression levels of Hes1 in brain AVMs [173]. Notch signaling is a key mechanotransduction pathway that controls cellular responses to mechanical cues in ECs [174]. In mouse models of AVM, increased arterial wall shear stress has been shown to activate the Notch1 and Notch4 signaling pathways in ECs [175]. Recently, Karthika et al. found that the intima of broad vessels in the cAVM nidus exhibited EndMT markers, such as SNAI1, SNAI2, calponin1, and transgelin [176]. Notch signaling plays a key role in cAVM pathogenesis, with oscillatory shear stress activating Notch3 expression in cerebral microvascular ECs. This shear-stress-activated Notch cascade increases N-cadherin expression levels and promotes EC infiltration. Human cAVMs exhibit lower levels of adhesion proteins, such as VE-cadherin and integrin α9/β1. In addition, deletion of the NOTCH pathway in pericytes reduces PDGFR-B expression levels and enhances apoptosis, indicating that NOTCH is essential for pericyte survival. In mice, the loss of vascular NOTCH signaling results in a reduction in the number of arterial pericytes and VSMCs, an increased number of VSMCs in veins, and acute AVMs in retinal arteries. Vascular abnormalities and pericyte loss have been observed in prenatal rat forebrains lacking NOTCH signaling [107].

## 5. Atherosclerosis and Notch Pathway

Vascular dysfunction and atherosclerosis are significantly associated with low shear stress. Atherosclerotic plaques usually develop at arterial bends and branching zones where disrupted flow causes low or oscillatory shear stress [177,178]. Alterations in signaling pathways, such as the Klf2/4 and Notch pathways, are typically associated with blood-flow-induced vascular dysfunction [179].

### 5.1. Shear Stress and Endothelial Dysfunction in Atherosclerosis

The accumulation of lipoproteins in the subendothelial region causes inflammation during atherosclerosis, which, in turn, causes endothelial dysfunction and adhesion molecule production. Notch signaling in ECs is disrupted by several factors, including inflammatory cytokines and altered shear stress [111,180,181]. Bifurcations and curvatures are areas of arteries with turbulent blood flow and disrupted shear stress, where atherosclerotic lesions are formed. Notch is a crucial transmitter of laminar shear stress [182]. Under conditions of low shear stress, reduced *miRNA126-5p* levels result in elevated Dll1 levels, which suppress Notch1 and impair the EC proliferation required for endothelial repair in dyslipidemia. Remarkably, *mir126*^(−/−)^ mice show more severe atherosclerosis [183]. In support of this, atheroprone areas of the mouse aorta show significantly decreased expression levels of Notch components [181]. Recently, Mack et al. explored the molecular mechanism underlying laminar shear stress-induced anti-atherogenic behaviors via the Notch pathway and found that a reduction in endothelial Notch1 levels causes atherosclerotic lesions in the descending aorta in the presence of hypercholesterolemia in mice [165]. Notch1 activation by shear stress is essential for maintaining junctional integrity, and its inhibition weakens endothelial junctions, allowing EC growth. Moreover, Polacheck et al. demonstrated that shear stress activates the Notch1 receptor, which is necessary for endothelial barrier function [118]. Shear stress causes the Dll4-dependent proteolytic activation of Notch1, thus exposing the transmembrane domain needed for endothelial junction construction. Miyagawa et al. revealed that the interaction between ECs and SMCs is vital for the activation of Notch1 through bone morphogenetic protein receptor 2 (BMPR2) [184]. In ECs, BMPR2 induces translocation of p-c-Jun N-terminal kinase to the membrane, thus stabilizing presenilin 1 and activating Notch1. Notch1 actively promotes EC proliferation, enhances glucose metabolism, improves mitochondrial function, and is essential for endothelial integrity and repair after injury.

### 5.2. Notch and Macrophages in Atherosclerosis

During the early stages of atherosclerosis, monocytes adhere to ECs, migrate to the intima, and differentiate into macrophages. As the disease progresses, inflammatory cytokines recruit additional monocytes, exacerbating plaque inflammation. M1 macrophages produce pro-inflammatory cytokines, such as IL-1 and TNF-α, whereas M2 macrophages release anti-inflammatory cytokines, such as IL-4, IL-10, and IL-13, to decrease inflammation [185]. Notch1 upregulates the production of IL-6, TNF-α, and MCP-1 in cultured monocytes, leading to M1 macrophage differentiation and inflammation. Conversely, Notch1 suppression promotes M2 differentiation by increasing the release of the anti-inflammatory cytokines IL-10 and IL-1RA [186,187]. Treatment of ApoE^−/−^ mice with DAPT causes reduced macrophage migration and repressed ICAM-1 expression in macrophages, which results in decreased macrophage infiltration in the atherosclerotic plaques [188]. Fung et al. presented the first direct evidence for the involvement of Notch in the regulation of macrophage function in atherosclerosis [189]. Notch3 and Dll4 are actively expressed in infiltrating macrophages and atherosclerotic plaques. In vitro, exposure to pro-inflammatory molecules, including IL-1β, LPS, and modified LDL, induces Dll4 expression in macrophages and triggers a cascade of pro-inflammatory responses, reinforcing a positive feedback loop in plaque-associated macrophages [189]. Pagie S et al. discovered that Dll4 expression increases in both ECs and macrophages during microvascular inflammation, indicating that it serves as a marker of endothelial activation and endothelial/macrophage interactions [190]. The same group established that Dll4 is a ligand involved in Notch-dependent M1 macrophage development, while inhibiting M2 differentiation and blocking IL-4-induced M2 gene expression. Fukuda et al. demonstrated that a high-cholesterol diet increased Dll4 expression levels in atherosclerotic plaques and adipose tissues [191]. Treatment with an anti-Dll4 antibody reduces plaque calcification, atherosclerotic lesions, fat formation, and insulin resistance. These outcomes are associated with decreased macrophage growth and MCP-1 levels. In vitro, Dll4-induced Notch signaling increases MCP-1 expression levels by activating NF-κB and promoting M1 macrophage polarization. Huang et al. confirmed that Notch controls M1/M2 polarization by demonstrating that *miR-148-3p*, produced after Notch1 activation, facilitates M1 polarization, while inhibiting M2 differentiation [192]. In ApoE^−/−^ mice, *miR-181b* regulates macrophage polarization by targeting Notch1. Moreover, Xu et al. demonstrated that Notch1 promotes mitochondrial glucose oxidation and activates the production of M1 pro-inflammatory genes [74]. One study showed that Notch regulates M2 macrophage polarization, with RBPJ-deficient mice demonstrating impaired M2 differentiation [71]. Ohishi et al. revealed that Dll1 inhibits the GM-CSF-driven differentiation of monocytes into macrophages, but promotes the differentiation of dendritic cells when GM-CSF, IL-4, and TNF-α are present [193]. Overall, these findings indicate that Notch signaling may serve as a potential target in various cell types to intervene in the progression of atherosclerosis. In addition, Notch signaling in vascular macrophages and monocytes contributes to inflammation by promoting the development of the pro-inflammatory M1 phenotype and suppressing the anti-inflammatory M2 subtype. Undoubtedly, the Dll4/Notch1 pathway is critical for this process, facilitating M1 differentiation and inhibiting M2 polarization.

### 5.3. Potential Therapeutic Targets in Atherosclerosis

There is emerging evidence for a clear link between disturbed Notch signaling and atherosclerosis, making both receptors and ligands promising therapeutic targets for inflammatory disorders. However, there are challenges in utilizing Notch for inflammation control while maintaining critical activities in tissue regeneration and homeostasis [194]. Notch signaling involves two primary strategies. First, γ-secretase inhibitors mainly suppress NCID release to decrease Notch levels and second, monoclonal antibodies (mAbs) can either suppress ligand binding or stabilize the NRR region to inhibit cleavage. Recent data demonstrated that most GSIs can inhibit atherosclerosis in ApoE^−/−^ mice by decreasing ICAM-1 expression levels and plaque formation [188]. Improved target selectivity, efficacy, and favorable dose intervals of mAbs would provide a more desirable option than pan-Notch inhibition. Experimental studies have provided strong evidence that targeting the Notch pathway may be helpful in the treatment of atherosclerosis. In a study of Ldlr^−/−^ mice, blocking DLL4-Notch signaling using an anti-DLL4 mAb decreased macrophage accumulation, insulin resistance, and plaque calcification and inhibited the progression of atherosclerosis [191].

## 6. Conclusions and Final Remarks

In this review, we present data demonstrating that the Notch pathway is essential for vascular development and influences processes that involve vasculogenesis and angiogenesis. It regulates the tip-to-stalk cell ratio in ECs to ensure a balance between sprouting, branching arterial-venous differentiation, and mural cell activity under various conditions. The lack of Notch pathway regulation contributes to a wide range of vascular diseases, emphasizing the potential for Notch-based treatments [195]. The Notch system provides context-specific outcomes through interactions with different signaling pathways, which are assisted by complicated regulatory mechanisms that require further investigation. The Notch pathway has been widely studied in relation to chronic and systemic inflammation and inflammatory diseases, and it is commonly believed that blocking Notch signaling has beneficial effects on inflammatory pathways [196]. Notch pathways influence vascular inflammation in a manner that is specific to both cell type and isoform. Targeting adequate Notch isoforms or Notch ligands has been proven to be an effective therapeutic strategy for vascular inflammatory diseases such as atherosclerosis. Immunotherapy in conjunction with Notch-targeted therapy requires additional efforts to elucidate the specific Notch receptors or ligands responsible for disease progression to minimize side effects and maximize treatment efficacy.

## Figures and Tables

**Figure 1 biomedicines-13-00837-f001:**
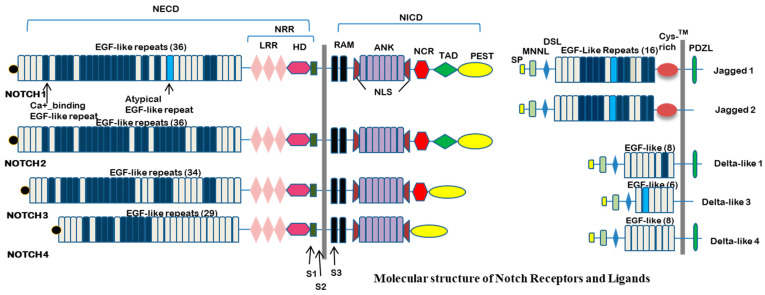
Molecular structure of Notch receptors and ligands: Mammals possess four notch receptors (Notch1-4) which belong to type I transmembrane proteins containing 29–36 epidermal growth factor (EGF) repeats, crucial for ligand binding. Adjacent to the EGF repeats, the NOTCH protein is organized with a negative regulatory region (NRR), containing 3 cysteine rich Lin repeats (LNR) and a heterodimerization domain (HD) which stabilizes the receptor before activation as well as intracellular components such as the RBP-Jk associated module (RAM) domain, nuclear localization sequences (NLS), seven ankyrin repeats (ANK), notch cytokine response (NCR) domain, transactivation domain (TAD), and regions rich in proline (P), glutamic acid (E), serine (S), and threonine (T) (PEST) domain. The Notch Extracellular Domain (NECD) and Notch Intracellular Domain (NICD) combine to produce a non-covalently coupled heterodimer. Ligand binding triggers sequential proteolysis (S1, S2, S3) by metalloproteases and γ-secretase, leading to NICD release. Notch ligands are categorized by the Jagged family (Jagged 1, Jagged 2), which contains a Module at the N-terminus of Notch Ligands (MNNL) domain, a Delta/Serrate/Lag-2 (DSL) domain, EGF repeats, and a cysteine-rich region, and the Delta-like family (DLL1, DLL3, DLL4), which lacks the cysteine-rich domain. Some ligands also contain an intracellular post-synaptic density protein ligand (PDZL) domain, which contributes to additional regulatory mechanism. The intracellular domain of ligands contains specific regions that facilitate endocytosis and regulate signal activity. Glycosylation is essential for both notch receptors and ligands for proper interaction and function.

**Table 1 biomedicines-13-00837-t001:** Genetic mouse models for targeted notch research.

Genetic Mouse Models for Targeted Notch Research			
**Components**	Whole Body	Endothelial Cell Specific	Smooth Muscle Cell Specific
Notch Receptors	N/A	*Notch 1 ^floxed-Tke2-Cre^* [37] *VE-Cadherin-Cre; Notch1^flox/flox^* [35] *Notch 2 ^ECKO-/-^* [65] *Notch 3 (N1IP::Cre^HI^)* [38] *Notch^+/-^Notch4^flf^* [66] *Prox1CreER^T2^* [67]	*Srf^flox/flox^* [68] *pdgfRβ-CreER^T2^* [69]
Notch Ligands	N/A	*Jag 1 ApoE^−/−^* [70] *Dll1^ECKO^* [71] *Dll4 ApoE^−/−^* [49]	*Myh11-CreER^T2^* [72] *pdgfRβ-CreER^T2^* [69]
RBPJ	*Rbpj^KO^*	*Cdh5(PAC)-CreER^T2^β* [73] *Rbpsuh (Rbpj)^flox^* [74] *Rosa26^mT/mG^* [75] *Rbpj^flox/flox^* [76]	*Rbpj-Smcko* [77]

**Table 2 biomedicines-13-00837-t002:** Notch-mediated endothelial regulation, with arrows denoting upregulation (↑) and downregulation (↓) of endothelial inflammation, vascular permeability and atherosclerosis.

Notch Components	Endothelial Inflammation	Vascular Permeability	Atherosclerosis
Notch 1	↑ [114,115]	↑	↑
	↓ [116,117]	↓	↓
Notch 2	↑ [112]		
Notch 3			
Notch 4	↑ [66,122]		
DLL1		↓ [123]	
DLL3			
DLL4			
Jag 1	↑ [70,117]		↑
Jag 2			

**Table 3 biomedicines-13-00837-t003:** Pathways mediating Notch-induced inflammatory signaling.

Pathways Mediating Notch-Induced Inflammatory Signaling	
NFκB	-Notch 1 interacts with p65 subunit of NF-κB to induce inflammatory cytokines expression such as Tnf-α and IL-1β expression [130] -TRAF6-IKK pathway regulation for enhancing TLR-4-induced pro-inflammatory cytokine expression [136]. -Notch 3 and Notch 1 activation by LPS increases NF-κB activity via p38 MAPK in macrohages [135]. -NF-κB upregulates Notch target genes like Hes1 [140]. -NF-κB suppresses Notch 4 and Hes1 expression in endothelial cells [43]
TLR	-TLR causes direct activation of notch target genes(Hes1and Hey1) in macrophages [131]. -TLR activation reduce IL-6 and TNFα production with NICD1 and NCID2 overexpression in macrophages [136]. -Notch signaling reduces TLR-mediated ERK phosphorylation and NF-κB activation [135].
HIF-1α	-HIF-1α and Notch1 ICD promotes expression of Hes1, Hey1, and Hey2 in endothelial cells [143]. -HIF-1α promotes stem cell diffenetiation by notch signaling [142]. -Activation of Notch signaling requires C-terminal of HIF-1α under inflammatory conditions [141]
Sirt1	-In Endothelial Cells, Sirt loss increases the expression of DLL4, NICD Hey1and Hes1, causing endothelial dysfunction, vascular leakage, and inflammation [68,126,127]. -In SMCs, Sirt overexpression enhances notch signaling to prevent vascular aging [68].
P13-AKT	-AKT activation enhances the notch signaling to promote cell migration and invasion in immune cells [25]. -Membrane-associated Notch can activate the PI3K-AKT pathway to increase the transcription of IL-10 and IL-12 [26]
Wnt	-Notch signaling modulates endothelial cell behavior, reducing Wnt-driven angiogenesis and inflammation [126]. -Notch signaling downregulate Wnt-induced stem cell proliferation, preventing excessive tissue repair and inflammation [144].

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
