# Peer review of "Multifaceted Role of Notch Signaling in Vascular Health and Diseases"

_biomedicines, 2025, doi:10.3390/biomedicines13040837_

Round 1
Reviewer 1 Report
Comments and Suggestions for Authors
The manuscript presents a comprehensive review of Notch signaling in vascular inflammation and mechanotransduction. It successfully integrates current knowledge of Notch-related molecular mechanisms with their implications in vascular biology and disease. The manuscript is well-structured, covering Notch’s role in vascular development, inflammation, and mechanotransduction.
If possible, include more statistical comparisons, effect sizes, and significance values when discussing experimental data.
Figure descriptions should include more details.
The review discusses various molecular pathways but does not include statistical data, meta-analyses, or quantitative comparisons of different studies.
The manuscript discusses macrophage polarization and Notch's role but does not provide quantitative data supporting these claims (for example: percentages of macrophage subtypes in atherosclerotic plaques).
Author Response
The manuscript presents a comprehensive review of Notch signaling in vascular inflammation and mechanotransduction. It successfully integrates current knowledge of Notch-related molecular mechanisms with their implications in vascular biology and disease. The manuscript is well-structured, covering Notch’s role in vascular development, inflammation, and mechanotransduction.
If possible, include more statistical comparisons, effect sizes, and significance values when discussing experimental data.
Figure descriptions should include more details.
Thank you. The figure legend has been more elaborated.
The review discusses various molecular pathways but does not include statistical data, meta-analyses, or quantitative comparisons of different studies.
Thank you for the comment. However, the purpose of our manuscript is not drawing meaningful conclusions based on meta-analysis, rather it is for reviewing and organizing recent finding related on our topic.
The manuscript discusses macrophage polarization and Notch's role but does not provide quantitative data supporting these claims (for example: percentages of macrophage subtypes in atherosclerotic plaques).
Thank you for the comment. However, we think providing individual data from the literatures we’ve cited is not critical and rather narrows our focus. I would appreciate your understanding.
Reviewer 2 Report
Comments and Suggestions for Authors
- While the current title is adequate, it does not fully capture the scope of this manuscript. Please consider revising the title to better reflect its content.
- Given the nature of this review paper, please consider modifying the table format for improved clarity and readability.
- This review paper discusses the multifaceted role of Notch signaling in vascular biology. However, the current conclusion focuses primarily on the role of Notch signaling in vascular development. Please expand the conclusion to include its relevance to atherosclerosis and inflammation.
- The authors have reviewed a substantial amount of information on Notch signaling activation and its potential targets. To enhance readability, it would be helpful to include a separate table summarizing these findings.
- It would be valuable to include a discussion on the available strains and models used to study Notch activation in the vascular microenvironment.
Author Response
While the current title is adequate, it does not fully capture the scope of this manuscript. Please consider revising the title to better reflect its content.
We’ve changed the title of the review.
Given the nature of this review paper, please consider modifying the table format for improved clarity and readability.
The table format has been modified.
This review paper discusses the multifaceted role of Notch signaling in vascular biology. However, the current conclusion focuses primarily on the role of Notch signaling in vascular development. Please expand the conclusion to include its relevance to atherosclerosis and inflammation.
Conclusion has been amended according to your suggestion.
The authors have reviewed a substantial amount of information on Notch signaling activation and its potential targets. To enhance readability, it would be helpful to include a separate table summarizing these findings.
We’ve prepared another table (table 3).
It would be valuable to include a discussion on the available strains and models used to study Notch activation in the vascular microenvironment.
We’ve made another table regarding mouse models used for Notch in vivo studies (table 1).
Reviewer 3 Report
Comments and Suggestions for Authors
A very useful and fairly detailed overview of the role of the Notch signaling pathway in vascular inflammation and mechanotransduction. The authors take a detailed look at the role of Notch in vascular development, homeostasis, inflammation, and mechanical signaling, making the review highly informative. It is gratifying that the role of Notch signaling is considered not only in the vascular, but also in the lymphatic system, which allows the reader to appreciate the versatility and significance of this signaling pathway. The advantages of the review include the fact that it considers the molecular aspects of Notch signaling.
I regret to note that the section linking Notch and vascular inflammation is written rather superficially. Since there is a lot of contradictory literature on this issue, I would consider that it is necessary not only to enumerate the various results, but also to critically analyze them. In the same section, I would consider it useful, firstly, to present more widely the data on the prospects of blocking Notch ligands for the control of inflammatory processes in blood vessels and, secondly, to focus on the fact that DLL4 blockade can be an effective way to treat cancer, due to the disruption of angiogenesis in the tumor.
Then, I would suggest that the authors carefully analyze the English text once again. In my opinion, there are some problems with the language here. For example, in the phrase "Shear stress is significantly associated with vascular dysfunction and atherosclerosis." (line 396) cause and effect are clearly confused; here it should be said that Vascular disfunction is significantly associated with low shear stress. I saw several such inaccuracies in the text: it is not difficult to eliminate them, but it is necessary.
Finally, while the authors have focused on recently published work on the role of Nitch signaling, I would advise them to pay more attention to the early published fundamental works relevant to the subject of the review, in particular “Notch1 Signaling Regulates Macrophage Polarization and Phenotype During Atherosclerosis Development.” by Cheng-Han Chen, et al., published in Arteriosclerosis, Thrombosis, and Vascular Biology in 2012.
In conclusion, the review makes a good impression and I think that it can be published after minor editing.
Comments on the Quality of English Language
I would suggest that the authors carefully analyze the English text once again. In my opinion, there are some problems with the language here. For example, in the phrase "Shear stress is significantly associated with vascular dysfunction and atherosclerosis." (line 396) cause and effect are clearly confused; here it should be said that Vascular disfunction is significantly associated with low shear stress. I saw several such inaccuracies in the text: it is not difficult to eliminate them, but it is necessary.
Author Response
A very useful and fairly detailed overview of the role of the Notch signaling pathway in vascular inflammation and mechanotransduction. The authors take a detailed look at the role of Notch in vascular development, homeostasis, inflammation, and mechanical signaling, making the review highly informative. It is gratifying that the role of Notch signaling is considered not only in the vascular, but also in the lymphatic system, which allows the reader to appreciate the versatility and significance of this signaling pathway. The advantages of the review include the fact that it considers the molecular aspects of Notch signaling.
Thanks a lot.
I regret to note that the section linking Notch and vascular inflammation is written rather superficially. Since there is a lot of contradictory literature on this issue, I would consider that it is necessary not only to enumerate the various results, but also to critically analyze them. In the same section, I would consider it useful, firstly, to present more widely the data on the prospects of blocking Notch ligands for the control of inflammatory processes in blood vessels and, secondly, to focus on the fact that DLL4 blockade can be an effective way to treat cancer, due to the disruption of angiogenesis in the tumor.
More experimental results regarding Notch ligands were added on page 7.
Then, I would suggest that the authors carefully analyze the English text once again. In my opinion, there are some problems with the language here. For example, in the phrase "Shear stress is significantly associated with vascular dysfunction and atherosclerosis." (line 396) cause and effect are clearly confused; here it should be said that Vascular disfunction is significantly associated with low shear stress. I saw several such inaccuracies in the text: it is not difficult to eliminate them, but it is necessary.
We’ve sent the manuscript for a professional English editing service.
Finally, while the authors have focused on recently published work on the role of Nitch signaling, I would advise them to pay more attention to the early published fundamental works relevant to the subject of the review, in particular “Notch1 Signaling Regulates Macrophage Polarization and Phenotype During Atherosclerosis Development.” by Cheng-Han Chen, et al., published in Arteriosclerosis, Thrombosis, and Vascular Biology in 2012.
Thank you but the paper you recommend can not be found on line.
In conclusion, the review makes a good impression and I think that it can be published after minor editing.